# Primary Efficacy of Co-Fumigation with Methyl Bromide and Phosphine Against *Tribolium castaneum* and *Sitophilus zeamais* in Wood Pellets as a Quarantine Treatment

**DOI:** 10.3390/insects16020186

**Published:** 2025-02-09

**Authors:** Donghun Cho, Min-Goo Park

**Affiliations:** 1Department of Plant Quarantine, Division of Pest Control, Animal and Plant Quarantine Agency (APQA), Gimcheon-si 39660, Republic of Korea; wasttime@korea.kr; 2Department of Bioenvironmental Chemistry, Jeonbuk National University, Jeonju 54896, Republic of Korea

**Keywords:** stored grain beetle, methyl bromide and phosphine mixture, quarantine treatment, sorption of fumigant, synergic effect

## Abstract

Pests found in wood pellets have been treated with methyl bromide (MB) fumigant, following Korea’s Phytosanitary Disinfection guidelines, which are intended for wood rather than wood pellets. MB fumigation results in low efficacy for the pests due to its high adsorption to the wood pellets, leading to repeated fumigation and increased logistics costs. This study aimed to evaluate the efficacy of using a combination of MB and phosphine (PH_3_) to disinfect stored product beetles in imported wood pellets. The adults and pupae of *Tribolium castaneum* and *Sitophylus zeamais* were selected as representative species and stages of invasive beetles, respectively. MB and PH_3_ exhibited a synergistic effect on both beetles. The study revealed that MB alone could not completely disinfect the two beetles owing to the high sorption to wood pellets, and the current 33 g/m3 MB treatment guideline for imported wood pellets in Korea is not optimal for achieving the target LCT_99_ level for the beetles. Our study provides an important insight into the combined MB + PH_3_, which is a potential alternative to the currently approved MB treatment for controlling exotic beetles in imported wood pellets in Korea.

## 1. Introduction

Wood pellets are biofuel produced by compressing wood fibers [1]. In 2021, the global wood pellets market was valued at USD 8.23 billion and is projected to increase at a compound annual growth rate of 5.5% from 2022 to 2030 [2]. Scientists and international organizations consider wood pellets to be a biofuel with potential climate benefits compared to fossil fuels. The increasing global demand for renewable energy drives the demand for wood pellets [3]. In 2022, several countries emerged as major import markets for wood pellets. The United Kingdom imported wood pellets valued at USD 1.6 billion, followed by Japan (USD 909.3 million), South Korea (USD 716.6 million), and Italy (USD 704.6 million) [4]. Korea imported 10.3 million tons of wood pellets between 2020 and 2022, averaging approximately 3.4 million tons per year [5]. When exotic pests are detected in wood pellets at Korean import ports, they undergo methyl bromide (MB) fumigation in accordance with the treatment guidelines of the Animal and Plant Quarantine Agency (APQA) [6]. Stored product beetles and ants such as *Carpophilus obsoletus*, *Tribolium castaneum*, *Cryptolestes ferrugineus*, and *Paratrechina longicornis* are frequently found in imported wood pellets despite the pellets being manufactured from wood fiber [5]. These species account for approximately 73% of the pests found in wood pellets between 2020 and 2022. Approximately 4% of the total imported volume, equivalent to 382 thousand tons of wood pellets, underwent fumigation owing to pest interceptions between 2020 and 2022 [5].

Although being phased out owing to its harmful effects on the ozone layer and human health, MB fumigation remains permissible for quarantine and pre-shipment (QPS) purposes under a critical use exemption granted by the United Nations Environmental Programme [7,8,9,10,11]. This is because effective alternative treatments for pest disinfection in many commodities are generally lacking, making MB essential to prevent the spread of pests between countries [8,12]. However, even when MB treatment is approved for QPS, its high sorption on certain commodities, such as wood pellets, can reduce its efficacy [13]. This underscores the need to develop alternative treatments to replace or reduce MB. The MB guideline, presently applied to imported wood pellets in Korea, was initially intended for wood, not wood pellets. The guideline specifies 33 g/m^3^ MB for 24 h at temperatures above 15 °C [6]. Processed wood pellets have altered physical properties, resulting in greater sorption of MB on wood pellets than on raw wood. This reduces the MB concentration during fumigation, requiring refumigation on the same commodity to entirely eliminate the pests [14].

Phosphine (PH_3_) is a traditional and effective alternative to MB for disinfecting stored products [15,16]. However, effective use of PH_3_ typically requires a long fumigation period (at least 48 h or more) [16], which may not be suitable for wood pellets that require faster treatment (within 24 h) to minimize logistics costs at the ports. An option to overcome the disadvantage of long-term fumigation is to combine fumigants such as MB and PH_3_. Previous studies indicated that the mixture of the two fumigants improved efficacy or reduced the fumigation period [17,18]. A new guideline is urgently needed to ensure a stable supply of wood pellets for electricity generation in Korea without adversely affecting climate change, although MB is included in the procedure.

*Tribolium castaneum* and *Sitophilus zeamais* are common stored product beetles that are found worldwide, including in Korea. *Tribolium castaneum* is relatively tolerant to MB, whereas *S. zeamais* shows tolerance to PH_3_ [13]. Therefore, these two beetles are suitable representatives of exotic beetles intercepted in Korea on which the efficacy of the two fumigants can be tested.

This study aimed to assess the potential of using MB and PH_3_ for disinfecting stored product beetles in imported wood pellets. The adults and pupae of *T. castaneum* and *S. zeamais* were chosen as representative species and stages of invasive beetles, respectively. The pupae exhibit the highest tolerance among the developmental stages of the two beetles [13]. The concentration of the combined MB and PH_3_ treatment required for 99% mortality (LC_99_) of *T. castaneum* and *S. zeamais* was determined. We (1) determined the sensitivity of the adult and pupae of *T. castaneum* and *S. zeamais* to MB and MB + PH_3_ at various temperatures; (2) examined the sorption of MB in different loading ratios of wood pellets at various temperatures and assessed the efficacy of each MB and PH_3_ on the beetles; and (3) evaluated the combined efficacy of MB and PH_3_ treatment on the pests at the highest loading ratio.

## 2. Materials and Methods

### 2.1. Insects

*Tribolium castaneum* and *S. zeamais* were provided by the Plant Quarantine Technology Center of APQA, Republic of Korea. The strains were PH_3_ resistant and originally purchased from Biological Utilization Institute (Andong, Republic of Korea). The stored grain beetles were reared in the laboratory at 26 ± 1 °C and 60–70% relative humidity with a 16:8 h (L:D) photoperiod. Flour (800 g), wheat bran (200 g), and dry yeast (70 g) were mixed to serve as a food source for *T. castaneum*, and brown rice was served as a food source for *S. zeamais*. The beetles were reared in plastic containers (20 cm W × 7 cm L × 8 cm H) for more than one year, indicating that the life cycle repeated more than 12 times for *S. zeamais* and 6 times for *T. castaneum* because one life cycle of *S. zeamais* and *T. castaneum* is approximately more than 30 days and 60 days, respectively.

In the colonies of *T. castaneum* and *S. zeamais* used in this experiment, males and females were distinguished based on morphological characteristics, such as sex patches. In both species, the female proportion was approximately 70%, while the male proportion was around 30%.

The ages of pupae and adults of *Tribolium castaneum* used in this study were approximately 55 days and 60 days, respectively. For *S. zeamais*, the ages of pupae and adults were approximately 23 days and 30 days, respectively.

*Tribolium castaneum* was collected based on visual assessment, while *S. zeamais* could not be collected this way because it commonly hides within its food source. To collect pupae of *S. zeamais*, adults of *S. zeamais* were reared on brown rice for 2 days, after which the brown rice and adults were separated. The eggs in the rice were allowed to develop, and after 23 days, the pupae were collected and used in the experiment.

The mortality of adults was assessed at the end of fumigation. Pupae were assessed 10 days after the end of fumigation to determine if emergence had occurred.

### 2.2. Fumigants

MB (purity: 98.0%) was supplied by the Animal and Plant Quarantine Agency (Gimcheon, Korea). PH_3_ (2% PH_3_ + 98% CO_2_) was purchased from Korea Nano Gas Co. (Yeoju, Korea).

### 2.3. Sensitivity Test for MB and MB + PH_3_ Against T. castaneum and S. zeamais

Adults and pupae of *T. castaneum and S. zeamais* were placed in separate Petri dishes (50 mm diameter × 15 mm height) containing flour or rice (20 pupae/dish and 20 adults/dish). Each dish containing pupae of *T. castaneum and S. zeamais* was fumigated with various dosages of MB and MB + PH_3_ in a 12 L desiccator for 24 and 48 h at different temperatures of 5, 10, and 20 °C to determine LC_50_ and LC_99_ values for the beetles. Adults of both beetle species were tested only for 24 h at 20 °C due to their high sensitivity to the fumigants. The temperature reflected the weather conditions from winter to summer in the Republic of Korea, when wood pellets are commonly imported. The dosage rates of MB + PH_3_ were applied at 100:1, 10:1, and 5:1. MB was injected through the inlet area using a gas-tight syringe (100 mL, Hamilton, NV, USA). For the MB + PH_3_ treatment, PH_3_ was administered concurrently with the application of MB. A mini-fan (6.5 cm i.d. × 3 cm) was placed at the bottom of the desiccator to promote air circulation. After fumigation, desiccators were opened and aerated for 2 h in a fume hood. The treated beetles were removed from desiccators, and adult mortality was assessed on the same day. The pupae were transferred to the rearing room (26 ± 1 °C and 60–70% relative humidity), and their mortality was determined by monitoring their emergence for 10 days. The mortality rate of the eggs of *S. zeamais* was calculated by comparing the average number of beetles in the untreated group with that in the treated groups. The trials were replicated thrice.

### 2.4. Efficacy on MB and PH_3_ Against T. castaneum and S. zeamais, Respectively, Based on the Dosage, Loading Ratio, Temperature, and Fumigation Time

The adults and pupae of *T. castaneum* were placed in separate rearing dishes (100 mm diameter × 40 mm height) containing flour or brown rice (50 pupae/dish and 50 adults/dish). Each dish was fumigated with 99–260 mg/L of MB in a 12 L desiccator with different loading ratios of 20, 40, and 60% based on weight per volume, such as 2.4 kg per 12 L, at temperatures of 5, 10, and 20 °C for 24 h. Wood pellets were incubated at temperatures of 5, 10, and 20 °C for 24 h, with the beetles being in the desiccator before fumigation. MB injection, circulation, and the process after completion of the trials followed the procedure described in Section 2.3. The trials were replicated 1–5 times to enhance experimental efficiency in determining the lethal doses of the pest.

The pupae and adults of *S. zeamais* were placed in separate rearing dish (100 mm diameter × 40 mm height) containing flour (50 pupae/dish and 100 adults/dish). Each dish containing pupae and adults of *S. zeamais* was fumigated with 2–4 mg/L of PH_3_ in a 12 L desiccator for 96–144 h at different temperatures of 5, 10, and 20 °C with the highest loading ratio of 60%. The injection and release of PH_3_ followed the same procedure as that for MB. The trials were replicated 1–3 times to enhance efficiency in determining the lethal doses of the pest.

### 2.5. Efficacy of MB + PH_3_ Against T. castaneum and S. zeamais at a Filling Ratio of 60% for a 24 h Fumigation

Pupae and adults of *T. castaneum* were placed in separate rearing dish (100 mm diameter × 40 mm height) containing flour or brown rice (50 pupae/dish and 50 adults/dish). Each dish was fumigated with 40 + 0.4 and 70 + 0.7 mg/L of MB + PH_3_ in a 12 L desiccator with a filling ratio of 60% at temperatures of 5, 10, and 20 °C for 24 h. The doses reflected the LCT_99_ value, which was obtained through sensitivity test of the beetles to MB or PH3 depending on the loading ratio according to Section 2.3 and Section 2.4. Preparation, treatment, and post-treatment followed the procedure in Section 2.3. The trials were replicated 1–3 times to enhance efficiency in determining the lethal doses of the pest.

For *S. zeamais*, pupae and adults were placed in each rearing dish (100 mm diameter × 40 mm height) containing flour (50 pupae/dish and 100 adults/dish). Each dish containing pupae and adults of *S. zeamais* was fumigated with 40 + 0.4 to 200 + 2.0 mg/L of MB + PH_3_ in a 12 L desiccator with a filling ratio of 60% at temperatures of 5, 10, and 20 °C for 24 h. The doses reflected the LCT_99_ value for achieving 100% mortality of the beetles. Preparation, treatment, and post-treatment followed the same procedure as that of MB alone. The trials were replicated 1–3 times to enhance efficiency in determining the lethal doses of the pest.

### 2.6. Determination of PH_3_ and MB Concentration and Concentration and Time (Ct) Product

To calculate the Ct product, the concentrations of MB and PH_3_ in the fumigation chambers were measured at 0.1, 1.0, 2.0, and 24 h or 0.1, 1.0, 2.0 h, 24 h, and 48 h after fumigant injection into chambers. The collected gas concentration was analyzed using gas chromatography (GC; 6890N and 7890A, Agilent Technology, Santa Clara, CA, USA). MB was detected using a flame ionization detector (FID), and the column was a DB5-MS (30 m × 0.25 mm i.d. × 0.25 µm film thickness; J&W Scientific, Folsom, CA, USA). The oven temperature was maintained constant at 100 °C, and injector and detector temperatures were 250 and 280 °C, respectively. Helium was used as a carrier gas at a flow rate of 1.5 mL/min. PH_3_ was detected using a nitrogen phosphorus detector (NPD). The GC conditions were as follows: GC NPD injector temperature (250 °C), oven temperature (240 °C), and detector temperature (320 °C), and the column was an HP-5 (0.53 mm × 15 m, Agilent Technology).

The concentrations of MB and PH_3_ were calculated based on peak areas using external standards. The calibration curve standards were prepared by spiking a known volume of MB and PH_3_ into a 1 l Tedlar^®^ gas sampling bag (SKC Inc., Eighty-four, PA, USA).

The Ct products were calculated based on the following equation as described by Ren et al. [19]:$$Ct=\sum \frac{\left(C_{i}+C_{i+1}\right)\left(t_{i+1}-t_{i}\right)}{2},$$
where C is the concentration of fumigant (mg/L), t is the time of exposure (h), i is the order of measurement, and Ct is the concentration × time product (g h/m^3^).

### 2.7. Data Analysis

The dose–response relations of *T. castaneum* and *S. zeamais* to MB and MB + PH_3_ were analyzed using Probit analysis [20], with a computer program developed by Ge Le Pattourel (Imperial College, London, UK) and utilized by Don-Pedro [21]. The analysis was conducted using SPSS (IBM, Version 22) without transforming the dose or CT product. In the Probit parameters, ‘df’ denotes the degrees of freedom related to the number of treated cases. LC_50_, LC_99,_ and LCT_99_ represent the concentration and Ct products at which 50% and 99% lethality of the pests are attained, respectively.

## 3. Results

### 3.1. Sensitivity Test for MB and MB + PH_3_ Against T. castaneum and S. zeamais

LC_50_ and LC_99_ of MB for adults and pupae of *T. castaneum* and *S. zeamais* increased as the temperature decreased for 24 and 48 h fumigation periods, as shown in Table 1 and Table 2. Pupae of both insects were more tolerant to MB than adults. LC_99_ of MB on *T. castaneum* pupae were 31.49, 39.21, and 53.48 mg/L at 20, 10, and 5 °C for 24 h, respectively, whereas the Ct products were 634.5, 810.0, and 1097.5 mg h/L (Table 1). LC_50_ and LC_99_ of MB + PH_3_ on adults and pupae of *T. castaneum* and *S. zeamais* increased as the temperature decreased for 24 and 48 h fumigation periods (Table 1 and Table 2). Pupae of both insects were more tolerant to MB + PH_3_ than adults. LC_99_ of *T. castaneum* pupae were 4.05 + 0.04, 6.08 + 0.06, and 20.85 + 0.21 mg/L for 24 h at 20, 10, at 5 °C, respectively, and the Ct products were 75.9 + 0.8, 111.2 + 1.2, and 400.6 + 3.8 mg h/L with a ratio of 100:1 for MB:PH_3_ (Table 1). This indicates that the LCT_99_ values were very high when MB was processed alone but significantly reduced when combined with PH_3_. The MB + PH_3_ with a ratio of 10:1 yielded similar results to that with a ratio of 100:1, indicating LCT_99_ values of 58.8 + 5.7 and 174.6 + 17.0 mg h/L at 10 and 5 °C, respectively.

The LC_99_ of MB on *S. zeamais* pupae were 24.32, 36.06, and 51.49 mg/L at 20, 10, and 5 °C, respectively, whereas the LCT_99_ were 502.1, 757.3, and 1098.9 mg h/L (Table 2). When MB was mixed with PH_3_, the LC_99_ of MB + PH_3_ on *S. zeamais* pupae were 4.68 + 0.05, 7.21 + 0.07, and 22.06 + 0.22 mg/L at 20, 10, and 5 °C, respectively, whereas the Ct products were 81.1 + 0.8, 122.1 + 1.2, and 455.7 + 4.6 mg h/L with a ratio of 100:1 for MB:PH_3_. This decreasing LCT_99_ pattern in the mixed treatment was similar to that for *T. castaneum*, including a ratio of 10:1. Details, including those for the Probit analysis, are presented in Table 2.

### 3.2. Efficacy of MB Against T. castaneum Based on the Dosage, Loading Ratio, and Temperature for 24 h Fumigation

The efficacy of MB against adults and pupae of *T. castaneum* decreased with a decrease in temperature and dosage and an increase in filling ratio, as presented in Table 3. The adults of *T. castaneum* exhibited increased sensitivity than the pupae. Adults were completely eradicated at concentrations of 128, 128, and 260 mg/L with a maximum filling ratio of 60% at 20, 10, and 5 °C, respectively. However, complete eradication of pupae (100%) was not achieved under the same conditions. The Ct product of MB on the treated groups, 161 and 128 mg/L, with a 20% filling ratio at 5 and 10 °C, and 161 mg/L, with a 60% filling ratio at 20 °C, showed high beetle mortality and did not achieve LCT_99_, as represented in Table 1. The result indicated that MB exhibited high sorption to wooden pellets.

### 3.3. Efficacy of PH_3_ Against S. zeamais Based on the Dosage, Temperature, and Time at a Loading Ratio of 60%

The efficacy of PH_3_ against adults and pupae of *S. zeamais* decreased with a decrease in temperature, time, and dosage of PH_3,_ as presented in Table 4. The adults of *S. zeamais* exhibited increased sensitivity compared to the pupae. At 5 °C, adults were completely eradicated, whereas pupae exhibited a mortality rate of 84.8%, corresponding to a corrected emergence rate of 15.2%. Considering the effect of temperature, treatment with 4 mg/L for 168 h at a temperature of 5 °C did not result in 100% mortality of *S. zeamais*. However, 100% disinfection was achieved with 4 mg/L for 144 h at 10 °C and with over 3 mg/L for 120 h at 20 °C for pupae.

### 3.4. Efficacy of MB + PH_3_ Against T. canstaneum and S. zeamais at a Filling Ratio of 60% for 24 h Fumigation

The combined treatment with MB and PH_3_ resulted in 100% mortality rates for *T. canstaneum* and *S. zeamais* after 24 h of fumigation at all temperatures at a 60% filling ratio, as provided in Table 5 and Table 6. A mixture of MB at 70 mg/L and PH_3_ at 0.7 mg/L for a 24 h fumigation period resulted in 100% mortality at 20 °C (Table 5), whereas 161 mg/L MB resulted in only 95.8% mortality, corresponding to a 4.2% corrected emergence rate, for *T. castaneum* pupae (Table 3). Complete disinfection of *S. zeamais* was not achieved with 4 mg/L of PH_3_ even after 96 h (Table 4). This indicates that the combined fumigation notably increased the efficacy of both MB and PH_3,_ consistent with the sensitivity tests (Table 1 and Table 2).

A higher concentration of MB + PH_3_ was required to disinfect *S. zeamais* compared to *T. castaneum* at temperatures ranging from 5 to 20 °C. MB mixed with PH_3_ at concentrations of 200 + 2.0, 160 + 1.6, and 100 + 1.0 mg/L achieved 100% mortality on *S. zeamais* at 5, 10, and 20 °C, respectively, whereas a concentration of only 70 + 0.7 mg/L thoroughly disinfected *T. castaneum* at the same temperatures.

## 4. Discussion

Our tests to evaluate the efficacy of MB and PH_3_ against *T. castaneum* and *S. zeamais* demonstrated that combining MB with PH_3_ notably reduced the required dosage of MB for pest disinfection. In a susceptibility test for *T. castaneum* pupae, the LC_99_ of MB was 31.49 mg/L at 20 °C. However, when 0.04 mg/L of PH_3_ was added, it significantly decreased to 4.05 mg/L (Table 1). Additionally, even at the highest filling ratio of 60% for the same pest, the combined treatment similarly reduced the lethal concentration of MB: Treatment with 161 mg/L of MB resulted in 95.8% mortality at 20 °C, whereas treatment with 70 + 0.7 mg/L MB + PH_3_ achieved 100% mortality (Table 3 and Table 5). The combined MB + PH_3_ treatment achieved complete mortality of adults and pupae of *T. castaneum* and *S. zeamais* at concentrations of 200 + 2.0, 160 + 1.6, and 100 + 1.0 mg/L after 24 h at temperatures of 5, 15, and 20 °C, respectively. Previous studies have also demonstrated the synergistic effect of the combined fumigation with MB + PH_3_ [22].

The efficacy of the fumigant is influenced by the fumigation temperature, species of pests, and the developmental stage of the pests. Lower fumigation temperatures require higher fumigant concentrations to control target pests [23,24,25]. Sulfuryl fluoride or ethyl formate requires high concentrations to disinfect fire ants or termites at lower temperatures. *Sitophilus zeamais* is more tolerant to PH_3_ than *T. castaneum.* Additionally, the pupae of both pests show higher tolerance to MB and PH_3_ than adults [13]. Our study also showed that the efficacy of MB and PH_3_ on *T. castaneum* and *S. zeamais* decreased as the fumigation temperature decreased. As the temperature decreased from 20 to 5 °C, the LC_99_ value for *T. castaneum* increased from 31.49 to 53.48 mg/L (Table 1), and it increased from 24.32 to 51.49 mg/L for *S. zeamais* (Table 2), showing a pattern consistent with previous studies. The LC_99_ of MB at 20 °C for *T. castaneum* pupae was 31.49 mg/L, which was higher than that for adults at 11.76 mg/L (Table 1), and the LC_99_ for *S. zeamais* pupae was 24.32 mg/L, which was higher than 11.32 mg/L for adult stages (Table 2). In addition, when PH_3_ was applied at 60%, a filling ratio similar to actual fumigation conditions, at 5 °C with a concentration of 4 mg/L for 168 h, adults were completely disinfected; however, pupae had a mortality of only 84.7%, showing a pattern identical to that reported in the previous studies. *Sitophilus zeamais* was more tolerant to the combined MB + PH_3_ treatment than *T. castaneum* due to differences in PH_3_ susceptibility between the two pests [13].

Fumigant adsorption varies by fumigant type and material. Grains with a smooth surface, such as rice or wheat, have low adsorption, whereas powdery, finely crushed products, such as pelletized feeds, exhibit very high adsorption [13]. The adsorption of MB is higher than that of PH_3_ but lower than that of sulfuryl fluoride or ethanedinitrile [19]. In this study, MB also showed high adsorption on wood pellets, a finely crushed product. The MB susceptibility test for *T. castaneum* at 20 °C showed a CT value of 634.5 mg h/L at an LC_99_ of 31.49 mg/L (Table 1); however, even at a concentration of 161 mg/L with a 60% filling ratio, only a CT value of 64.2 mg h/L was observed, which was approximately 1/10 (Table 3). This resulted in only 90% of the target pests being eradicated using MB alone at the actual filling ratio. This indicates that the disinfection treatment standard of 33 g/m^3^ used for wood over 20 °C in Korea is inappropriate [6].

Most of the stored product beetles intercepted from wood pellets in Korea belong to the genera *Carpophilus*, *Cryptolestes*, and *Tribolium* [5]. The beetle species tested in this study, *T. castaneum* and *S. zeamais*, are established exotic species in Korea and were selected as representative pests for conducting this study owing to their tolerance to MB or PH_3_. A limitation of our study is that we focused solely on testing the adults and pupae of *T. canstaneum* and *S. zeamais*. Therefore, it is unclear whether other life stages of the two beetles and, more importantly, other exotic species of stored product beetles intercepted from imported wood pellets can be effectively treated by MB + PH_3_. Some trials in this study were also limited by the number of replicates or the high population density of beetles. Consequently, further studies are needed to test the efficacy of MB + PH_3_ against beetle eggs and larvae, including confirmatory trials using imported wood pellets infested with a wider range of exotic stored grain beetles in large fumigation enclosures such as 40-foot containers. Moreover, a larger number of beetles (e.g., 30,000 individuals) in the stage most tolerant to MB + PH_3_ should be tested in confirmation trials to minimize the impact of the beetles’ sex on mortality.

Nevertheless, this study provides several critical insights: (1) MB and PH_3_ exhibited a synergistic effect on both beetles; (2) MB alone could not completely disinfect the two beetles owing to the high sorption to wood pellets; and (3) the current MB treatment protocol of 33 g/m^3^ MB for imported wood pellets in Korea is not optimal for achieving the target LCT_99_ required for effective beetle management, highlighting the need for a reassessment and potential revision of the established treatment guidelines (200 + 2.0, 160 + 1.6, and 100 + 1.0 mg/L of MB + PH_3_ at 5, 10, and 20 °C).

In conclusion, MB + PH_3_ treatment represents a promising alternative to the currently approved MB treatment for the control of exotic beetles in imported wood pellets in Korea.

## Figures and Tables

**Table 1 insects-16-00186-t001:** Probit analysis of the efficacy of methyl bromide against *Tribolium castaneum* was conducted for 24 and 48 h fumigation periods at temperatures of 20, 10, and 5 °C.

Fumigant		Stage	Treatment		Lethal Concentration (mg/L)		df	*p*-Value	LCT_99_ (mg·h·L^−3^)
			Time	Temp	LC_50_ (95% CL)	LC_99_ (95% CL)			
			(h)	(°C)					
MB		Adult	24	20	10.66	11.76	3	<0.0001	-
					(10.60–10.73)	(11.60–11.98)			
		Pupa			27.40	31.49	7	<0.0001	634.5
					(25.73–31.56)	(28.65–39.31)			
				10	32.82	39.21	3	<0.0001	810.0
					(30.02–39.15)	(34.78–51.21)			
				5	45.59	53.48	4	<0.0001	1097.5
					(43.38–49.05)	(49.82–60.00)			
MB +PH_3_ (100:1)	MB	Adult		20	1.95	2.34	2	<0.0001	-
					(1.80–2.18)	(2.13–2.68)			
	PH_3_				0.02	0.02	2	<0.0001	-
					(0.02–0.02)	(0.02–0.03)			
	MB	Pupa			2.70	4.05	4	<0.0001	75.9
					(2.39–3.15)	(3.51–4.97)			
	PH_3_				0.03	0.04	4	<0.0001	0.8
					(0.02–0.03)	(0.04–0.05)			
	MB			10	2.34	6.08	4	<0.0001	111.2
					(1.89–2.67)	(5.47–7.03)			
	PH_3_				0.02	0.06	4	<0.0001	1.2
					(0.02–0.03)	(0.06–0.07)			
	MB			5	17.25	20.85	3	<0.0001	400.6
					(16.49–18.36)	(19.49–23.06)			
	PH_3_				0.17	0.21	3	<0.0001	3.8
					(0.17–0.18)	(0.19–0.23)			
MB +PH_3_ (10:1)	MB				6.25	8.54	2	<0.0001	174.6
					(5.71–6.98)	(7.69–9.82)			
	PH_3_				0.63	0.85	2	<0.0001	17.0
					(0.57–0.70)	(0.77–0.98)			
	MB			10	2.06	3.16	2	<0.0001	58.8
					(1.78–2.54)	(2.65–4.28)			
	PH_3_				0.21	0.32	2	<0.0001	5.7
					(0.18–0.25)	(0.26–0.43)			
	MB		48	5	2.45	3.33	2	<0.0001	110.0
					(2.21–2.81)	(2.94–3.97)			
	PH_3_				0.24	0.33	2	<0.0001	10.7
					(0.22–0.28)	(0.29–0.40)			

- The CT product for the adult stage was not calculated due to the high sensitivity of the beetles to methyl bromide.

**Table 2 insects-16-00186-t002:** Probit analysis of the efficacy of methyl bromide against *Sitophilus zeamais* was conducted for 24 and 48 h fumigation periods at temperatures of 20, 10, and 5 °C.

Fumigants		Stage	Treatment		Lethal Concentration (mg/L)		df	*p*-Value	LCT_99_ (mg·h·L^−3^)
			Time (h)	Temp (°C)	LC_50_ (95% CL)	LC_99_ (95% CL)			
MB		Adult	24	20	8.09 (7.31–8.80)	11.32 (10.15–14.53)	3	<0.0001	-
		Pupa			17.94 (17.08–18.62)	24.32 (22.90–26.84)	6	<0.0001	502.1
				10	23.51 (22.63–24.29)	36.06 (34.32–38.47)	4	<0.0001	757.3
				5	33.81 (32.66–34.80)	51.49 (49.47–54.14)	4	<0.0001	1098.9
MB +PH_3_ (100:1)	MB	Adult		20	1.61 (1.43–1.77)	2.99 (2.67–3.52)	3	<0.0001	-
	PH_3_				0.02 (0.01–0.02)	0.03 (0.03–0.04)	3	<0.0001	-
	MB	Pupa			1.77 (1.34–2.10)	4.68 (4.02–5.88)	4	<0.0001	81.1
	PH_3_				0.02 (0.01–0.02)	0.05 (0.04–0.06)	4	<0.0001	0.8
	MB			10	2.87 (2.26–4.83)	7.21 (6.76–7.90)	4	<0.0001	122.1
	PH_3_				0.03 (0.02–0.05)	0.07 (0.07–0.08)	4	<0.0001	1.2
	MB			5	11.88 (9.52–13.16)	22.06 (19.95–26.49)	5	<0.0001	455.7
	PH_3_				0.12 (0.95–0.13)	0.22 (0.20–0.26)	5	<0.0001	4.6
MB +PH_3_ (10:1)	MB				2.50 (0–4.23)	15.95 (12.14–29.14)	3	<0.0001	325.1
	PH_3_				0.25 (0–0.42)	1.60 (1.21–2.91)	3	<0.0001	33.1
	MB			10	3.06 (2.41–3.43)	4.63 (3.99–6.11)	3	<0.0001	71.2
	PH_3_				0.31 (0.24–0.34)	0.46 (0.40–0.61)	3	<0.0001	7.1
	MB		48	5	0.80 (0–1.30)	4.20 (3.38–6.32)	3	<0.0001	144.1
	PH_3_				0.080 (0–0.13)	0.420 (0.34–0.63)	3	<0.0001	14.8
MB +PH_3_ (5:1)	MB		24	10	0.64 (0–1.22)	4.38 (3.54–6.45)	3	<0.0001	65.0
	PH_3_				0.13 (0–0.24)	0.88 (0.71–1.29)	3	<0.0001	13.3
	MB		48		0.80 (0–1.26)	3.80 (3.07–5.68)	3	<0.0001	77.7
	PH_3_				0.16 (0–0.25)	0.76 (0.61–1.14)	3	<0.0001	15.0

- The CT product on adult stage were not calculated due to the high sensitivity of the beetle to methyl bromide.

**Table 3 insects-16-00186-t003:** Efficacy of methyl bromide against *Tribolium castaneum* based on filling ratios and dosage during 24 h fumigation.

Temp. (°C)	Filling Ratio (%)	MB (mg/L)	No. of Replication	Adults			Pupae			
				Total No.	Dead No.	Mortality (%)	Total No.	Emergence Rate (%)	Corrected Emergence Rate (%) ^1^	CTP (mg·h·L^−3^)
5	0	control	3	150	0	0	150	83.3 ± 7.0	-	-
	60	128	1	-	-	-	50	62.0	74.5	-
		161	1	-	-	-	50	50.0	60.0	-
		194	1	-	-	-	50	26.0	31.2	-
		260	2	100	100	100	100	8.0 ± 11.3	9.6 ± 13.6	-
	40	161	1	50	6	12	50	16.0	21.0	-
		194	1	-	-	-	50	4.0	4.8	-
		227	2	-	-	-	100	3.0 ± 4.2	3.6 ± 5.1	-
		260	5	250	250	100	250	2.0 ± 2.8	2.4 ± 3.4	186.6 ± 6.5
	20	128	1	-	-	-	100	4.0	4.8	-
		161	5	-	-	-	250	1.6 ± 3.6	1.9 ± 4.3	303.9 ± 18.1
10	0	control	3	150	0	0	150	90.0 ± 2.0	-	-
	60	128	1	50	50	100	50	36.0	40.0	-
		161	1	50	50	100	50	2.0	2.2	-
		194	1	-	-	-	50	52.0	57.7	-
		227	5	250	250	250	250	1.6 ± 2.6	1.8 ± 2.9	79.5 ± 9.2
	40	99	1	-	-	-	50	26.0	28.8	-
		128	5	250	250	100	250	2.0 ± 4.5	2.2 ± 5.0	129.3 ± 20.7
	20	33	1	50	6	88.0	50	82.0	91.1	-
		66	1	50	50	100	50	2.0	2.2	-
		99	5	250	250	100	250	1.2 ± 1.8	1.3 ± 2.0	65.3 ± 3.7
20	0	control	3	150	0	0	150	94.7 ± 1.2	-	-
	60	99	1	50	48	96.0	50	16.0	16.9	-
		128	5	250	250	100	250	3.6 ± 5.0	3.8 ± 5.2	54.9 ± 4.6
		161	3	-	-	-	150	4.0 ± 4.0	4.2 ± 4.2	64.2 ± 4.6

^1^ Corrected emergency rate = (1 − [mean number of emergence in control group − mean number of emergence in treated group]/mean number of emergence in control group) × 100. Mortality is equal to 100 corrected emergency rate. - Data could not be obtained because the experimental resources were limited, and the trials focused on highly tolerant stages.

**Table 4 insects-16-00186-t004:** Efficacy test of PH_3_ against *Sitophilus zeamais* based on dosage and exposure time at a filling ratio of 60%.

Temp. (°C)	Time (h)	PH_3_ (mg/L)	No. of Replication	Adults			Pupae		
				Total No.	Dead No.	Mortality (%)	Total No.	Emergence Rate (%)	Corrected Emergence Rate (%) ^1^
5	144	control	1	100	2	2	50	15	-
		4	1	100	97	97	50	23	69.6
	168	control	1	100	3	3	50	13	-
		4	3	300	300	100	150	4.0 ± 4.0	15.2 ± 15.4
10	96	control	1	100	25	25	-	-	-
		4	1	100	92	92	-	-	-
	120	control	1	100	44	44	-	-	-
		2	1	100	82	82	-	-	-
		3	1	100	95	95	-	-	-
		4	1	100	99	99	-	-	-
	144	control	1	100	5	5	50	28	56
		4	3	300	300	100	150	0	0
20	96	control	3	100	0	0	50	88	-
		2	1	100	76	76	-	-	-
		3	2	200	198	99.0 ± 1.4	-	-	-
		4	3	300	299	99.7 ± 0.6	50^2^	12	13.6
	120	control	1	100	0	0	50	76	-
		2	1	100	100	100	-	-	-
		3	3	300	300	100	150	0	0
		4	1	100	100	100	50	0	0

^1^ Corrected emergency rate = (1 − [mean number of emergence in control group − mean number of emergence in treated group]/mean number of emergence in control group) × 100. Mortality is equal to 100 corrected emergency rate. - Data could not be obtained because the experimental resources were limited, and the trials focused on higher doses and longer fumigation periods of PH_3_.

**Table 5 insects-16-00186-t005:** Efficacy test of MB + PH_3_ against *T. castaneum* at a filling ratio of 60% for 24 h fumigation.

Temp. (°C)	MB + PH_3_ (mg/L)	No. of Replication	Adult			Pupa			
			Total No.	Dead No.	Mortality (%)	Total No.	Alive No.	Emergency Rate (%)	Corrected Emergency Rate (%) ^1^
5	control	3	150	0	0	150	125	83.3 ± 7.0	-
	40 + 0.4	1	50	50	100	50	1	2.0	2.4
	70 + 0.7	3	150	150	100	150	0	0.0	0.0
10	control	3	150	0	0	150	135	90.0 ± 2.0	-
	40 + 0.4	1	50	49	98.0	50	14	28.0	31.1
	70 + 0.7	3	150	150	100	150	0	0.0	0.0
20	control	3	150	0	0	150	142	94.7 ± 1.2	-
	40 + 0.4	1	50	10	80	50	47	94.0	99.3
	70 + 0.7	3	150	150	100	150	0	0.0	0.0

^1^ Corrected emergency rate = (1 − [mean number of emergence in control group − mean number of emergence in treated group]/mean number of emergence in control group) × 100. Mortality is equal to 100 corrected emergency rate.

**Table 6 insects-16-00186-t006:** Efficacy test of MB + PH_3_ against *S. zeamais* at a filling ratio of 60% for 24 h fumigation.

Temp. (°C)	MB + PH_3_ (mg/L)	No. of Replication	Adult			Pupa			
			Total No.	Dead No.	Mortality (%)	Total No.	Alive No.	Emergency Rate (%)	Corrected Emergency Rate (%) ^1^
5	control	3	300	0	0	150	106	70.7 ± 3.1	-
	160 + 1.6	2	200	198	100	100	1	1.0 ± 1.4	1.4 ± 2.0
	200 + 2.0	3	300	300	100	150	0	0.0	0.0
10	control	3	300	0	0	150	125	83.3 ± 1.2	-
	100 + 1.0	1	100	99	99	50	8	16.0	19.2
	130 + 1.3	1	100	99	99	50	10	20	24.0
	160 + 1.6	3	300	300	100	150	0	0.0	0.0
20	control	mean	300	0	0	150	137	91.3 ± 4.2	-
	40 + 0.4	1	100	89	89	50	5	10.0	10.9
	70 + 0.7	3	300	300	100	150	8	5.3 ± 5.0	5.8 ± 5.5
	100 + 1.0	3	300	300	100	150	0	0.0	0.0

^1^ Corrected emergency rate = (1 – [mean number of emergence in control group − mean number of emergence in treated group]/mean number of emergence in control group) × 100. Mortality is equal to 100 corrected emergency rate.

## Data Availability

All data supporting the findings of this study are available from the corresponding authors upon reasonable request.

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
