# Peer review of "Primary Efficacy of Co-Fumigation with Methyl Bromide and Phosphine Against Tribolium castaneum and Sitophilus zeamais in Wood Pellets as a Quarantine Treatment"

_insects, 2025, doi:10.3390/insects16020186_

Round 1

Reviewer 1 Report

Comments and Suggestions for Authors

The paper considers an interesting and widespread practical aspect of trade. Pellets construction materials greatly affect such a situation because of their ability to harbor pest animals (Insects and Rodents) that are passively transported to various parts of the planet. The species examined in the manuscript are two Tribolium castaneum and Sitophilus zeamais. The former species is very recurrent being detrital, while the interception of the latter is rather rare being a primary internal feeder pest especially of grain

The experimental tests performed were done under controlled laboratory conditions and consider adults and pupae of the two species. Some important details to note are not clear from the text: the age of the specimens used, their sex, and for Sitophilus how the authors selected pupae that live well hidden in the food substrate. The results reported by the authors are closely related to such details that need to be carefully specified in the paper

My advice is to carefully revise the manuscript by adding and editing the relevant parts. In the attached PDF I have highlighted some parts that need revision and included notes that need explanations.

Reviewer 2 Report

Comments and Suggestions for Authors
  1. The title should specify "primary efficacy" as the study only tested pupae and adults. Additionally, "co-fumigation" is more appropriate for describing the mixed use of methyl bromide and phosphine.
  2. Replace "EF" with "PH₃" in Line 166.
  3. In Section 2.6 on probit analysis, the model used should be explicitly described, and it should be clarified whether the dose was transformed during the analysis.
  4. From Table 1, the LCT₉₉ value for methyl bromide fumigation is almost equivalent to the minimum concentration-time (CT) required for the phytosanitary treatment of wood packaging material under ISPM 15. This suggests that stored insects in vitro exhibit high resistance to methyl bromide. However, as the estimated LCT might decrease if the dose is not transformed, it is recommended that the authors explore alternative models for the probit analysis to validate these results.
  5. For the LC values presented, only lower and upper confidence intervals are provided. Please revise the word mix.-mix. for clarity and completeness.
  6. The current table formatting may confuse readers. It is suggested to standardize the data presentation and maintain two decimal places consistently across all tables.
  7. The limitations or shortcomings of this investigation should be discussed more thoroughly:
    • Resistance life stages: The most resistant life stages of these two insect species should be identified and tested.
    • Temperature conditions: Why were tests conducted at 5℃ but not at 15℃?
    • High MB doses: Clarify the rationale for using high methyl bromide doses (≥70 mg/L) in the preliminary dose-response tests.

Reviewer 3 Report

Comments and Suggestions for Authors

Disinfesting quarantine commodities is critical to prevent the entry of exotic pests and the paper presents an interesting topic on combining MB and PH3 for quarantine purposes. But the manuscript is deficient on the following aspects

Line 33 – Modify ‘for’ with ‘against’

Line 36 – Both pests or both stages

Abstract lacks information about the results of sorption studies and the effect of fumigants on pests at the highest loading ratio

Line 134 – Explain the reason for taking 100 adults in a 50 mm x 15 mm Petri dish. 100 adults/ Petri dish leads to crowding of test insects resulting in stress (Ramadan et al., 2024)

Section 2.3 mentions about both pupal and adult stages of test insects bioassayed at temperatures of 5, 10 and 20°Celsius, but Table 1 and 2 gives the results for adults only at 20°C and not at lower temperatures.

Section 2.4 mentions about testing MB and PH3 against rice weevil and red flour beetle. But Tribolium is tested only against MB Sitophilus against PH3.

Table 1 and 2 looks extremely confusing. The adult stages, various treatment duration and various treatment temperatures are missing in the table. The reason for omitting the data should be clearly stated.

In table 3, the unit of MB should be mentioned

In table 3 and 4 several data points are missing. In table 3, its with adults and its with pupa. The materials and methods and results sections does not mention anything about these omissions. The lines 229-231 states that adults were completely eradicated at doses 128 (repeated twice) and 260 mg/L, but the table does not show any values for this.

The unit of time in table 4 is missing

The discussion section is inadequate.

Round 2

Reviewer 1 Report

Comments and Suggestions for Authors

The Authors of the manuscript tried to respond to my previous comments, however their arguments were not very satisfactory. In my opinion the work is interesting because it examines a topic which from a practical operational point of view poses important general problems. For this reason I still ask the Authors to respond with greater attention to my previous comments.

Reviewer 2 Report

Comments and Suggestions for Authors

1.      Please clarify whether the CT or concentration was adjusted/transformed during the probit analysis.

2.      Replace "adult" with "adults" and "pupa" with "pupae" in Tables 3 and 4, as they refer to multiple individuals.

Reviewer 3 Report

Comments and Suggestions for Authors

The authors have incorporated most of the suggestions in the first round of review.

The authors have corrected the size of the Petri plate in the revised manuscript. In a 100 mm Petri plate also 100 adult insects is extremely high for a valid bioassay

Authors are requested to consider repeating the bioassays with a maximum of 20 adults

Round 3

Reviewer 1 Report

Comments and Suggestions for Authors

The Authors of the manuscript have partly justified my comments made in the first and second revisions. Overall, the new version of the work is quite satisfactory and can be considered for publication especially because of the rather innovative topic.

Reviewer 3 Report

Comments and Suggestions for Authors

The results of the bioassays with extremely high test insects in a Petri plate will not give a valid result. Hence I am constrained to accept the paper in its present form